# Efficacy of vonoprazan against bleeding from endoscopic submucosal dissection-induced gastric ulcers under antithrombotic medication: A cross-design synthesis of randomized and observational studies

Yu Hidaka[1]*, Toru Imai[2], Tomoki Inaba[3], Tomo Kagawa[3], Katsuhiro Omae[4], Shiro Tanaka[2]

1 Department of Biomedical Statistics and Bioinformatics, Kyoto University Graduate School of Medicine, Kyoto, Japan, 2 Department of Clinical Biostatistics, Kyoto University Graduate School of Medicine, Kyoto, Japan, 3 Department of Gastroenterology, Kagawa Prefectural Central Hospital, Takamatsu, Kagawa, Japan, 4 Department of Data Science, National Cerebral and Cardiovascular Center, Suita, Osaka, Japan

* yhidaka@kuhp.kyoto-u.ac.jp

**Data Availability Statement:** Given the use of human research participants' data in this study,

## Abstract

Vonoprazan, a potassium-competitive acid blocker, is expected to be superior to proton pump inhibitors (PPIs) in preventing post-endoscopic submucosal dissection (ESD)-induced gastric bleeding. However, the results of randomized controlled trials (RCTs) and observational studies on the efficacy of vonoprazan have been inconsistent. This study aimed to evaluate the effectiveness of vonoprazan in antithrombotic drug users, a population that has been excluded from RCTs. Treatment effects were assessed using cross-design synthesis, which can be adjusted for differences in study design and patient characteristics. We used data from an RCT in Japan (70 patients in the vonoprazan group and 69 in the PPI group) and an observational study (408 patients in the vonoprazan group and 870 in the PPI group). After matching, among the antithrombotic drug users in the observational study, post-ESD bleeding was noted in 8 out of 86 patients in the vonoprazan group and 18 out of 86 patients in the PPI group. After pooling the data from the RCT and observational study, the risk difference for antithrombotic drug users was -14.6% (95% CI: -22.0 to -7.2). CDS analysis suggested that vonoprazan is more effective than PPIs in preventing post-ESD bleeding among patients administered antithrombotic medications.

## Introduction

Endoscopic submucosal dissection (ESD) of gastric neoplasms is a minimally invasive and well-established procedure [1]. However, post-ESD bleeding from artificial ulcers remains an unsolved problem [2–4]. Vonoprazan (Takeda Pharmaceutical Co., Ltd., Tokyo, Japan) directly inhibits proton pumps without requiring activation by gastric acid. In addition, it

disclosing further information may reveal sensitive patient information and cannot be shared in the open domain. Data is available from the Ethics Committee of Kagawa Prefectural Central Hospital (contact via their website; http://www.chp-kagawa.jp/) for researchers who meet the criteria to access confidential data.

**Funding:** This study was funded by AMED under Grant Numbers JP19lk0201061 and JP20lk0201061 and 21lk0201702. The funders had no role in study design, data collection and analysis, decision to publish, or preparation of the manuscript.

**Competing interests:** The authors have declared that no competing interests exist.

rapidly and strongly suppresses acid secretion [5]. Vonoprazan has a dissociation coefficient of 9.37 and is more likely to accumulate in the secretory tubules of the stomach than proton pump inhibitors (PPIs) [6]. In addition, vonoprazan offers several advantages over PPIs that allow it to achieve a higher gastric pH. These include rapid drug absorption after administration (median Tmax, 2 h), longer plasma half-life than PPIs, and no interindividual pharmacokinetic variations due to differences in cytochrome *P450 2C19* (*CYP2C19*) genotypes [7]. A crossover study demonstrated that vonoprazan could suppress gastric acid secretion rapidly and more strongly than PPIs [8]. Therefore, vonoprazan is expected to be more effective than PPIs for treating artificial gastric ulcers resulting from ESD. Our previous observational study showed that vonoprazan suppressed post-ESD bleeding more effectively than PPIs [9]. However, subsequent randomized controlled trials (RCTs) comparing PPIs and vonoprazan in terms of post-ESD bleeding control did not find vonoprazan to be superior to PPIs [10–15].

With an increase in the aging population, the number of patients being administered antithrombotic drugs is increasing in developed countries. The proportion of patients undergoing gastric ESD who receive antithrombotic drugs is also expected to increase [16]. A meta-analysis revealed that antithrombotic drug users have a significantly increased risk of post-ESD bleeding than non-users, regardless of whether they suspended the treatment perioperatively. However, the analysis only included studies in which PPIs were used to treat post-ESD ulcers [17]. To improve the safety of gastric ESD, it is important to determine whether vonoprazan suppresses post-ESD bleeding more effectively than PPIs in individuals consuming antithrombotic drugs.

A subgroup analysis in our observational study suggested that vonoprazan suppresses post-ESD bleeding in antithrombotic drug users [9]. Safety-critical RCTs generally exclude patients at high risk of trial-related complications. Indeed, RCTs with post-ESD bleeding as the primary endpoint excluded antithrombotic drug users [10, 11]. In RCTs that included antithrombotic drug users, it was difficult to draw firm conclusions regarding the prevention of post-ESD bleeding for several reasons. These reasons include the primary endpoint being ulcer reduction, small sample size, inclusion sample size of antithrombotic drug users, low post-bleeding rates, and slow initiation of vonoprazan [12–15].

In real-world clinical practice, complications associated with ESD have major consequences in elderly individuals and patients with serious comorbidities who cannot be included in RCTs. Cross-design synthesis (CDS) is a method for estimating the treatment effect in patients excluded from RCTs. It involves pooling the data of an observational study and an RCT to estimate the treatment effect in the overall treated population [18, 19]. In the present study, we used CDS to evaluate the inhibitory effect of vonoprazan on post-ESD gastric ulcer bleeding in patients consuming antithrombotic drugs.

## Methods

### Study design and data collection

In this study, we used CDS, a method that pools the results of an observational study and an RCT, to estimate treatment effects in patients excluded from the RCT and the overall treated population. Specifically, the observational study included subgroups that met the exclusion criteria of the RCT. The CDS method, designed to address the generalizability problem in a real-world clinical setting caused by sample selection bias in RCTs, is used to estimate the effect of a treatment on a patient population that is unable to participate in RCTs using observational data. As the patients in the observational study were not randomized, the results could be biased due to treatment selection errors. However, the CDS estimator is unbiased if the assumption that the treatment selection error for stratified estimators from the observational

**Table 1. Summary of pooled studies.**

| | RCT (Hamada et al., 2019) | Observational study |
|---|---|---|
| Study design | Single-centre, randomized phase II trial | Retrospective observational cohort study |
| Study period | 2015–2016 | 2005–2018 |
| Research facilities | Osaka International Cancer Institute | Kagawa Prefectural Central Hospital |
| Inclusion criteria | (i) Age 20 years or older<br>(ii) ECOG performance status 0–2<br>(iii) Hb≥9 g/dl; Plt≥100,000/ mm 3; AST, ALT≤100 U/l; Cre≤2.0 mg/dl; PT≥70%. | Age 20 years or older |
| Exclusion criteria | (i) Concurrent endoscopic treatment for esophageal or duodenal lesion<br>(ii) History of endoscopic treatment in the past 28 days<br>(iii) Endoscopic treatment scheduled within 28 days after gastric ESD<br>(iv) History of gastric resection<br>(v) allergy to vonoprazan or lansoprazole<br>(vi) Systemic administration of corticosteroids, anticoagulant agents, or antiplatelet agents<br>(vii) Systemic administration of nonsteroidal anti-inflammatory drugs that could not be suspended between 7 days before gastric ESD and 28 days afterwards<br>(vii) Pregnancy or lactation<br>(ix) Major organ failure | (i) Remnant stomach<br>(ii) Surgical resection of the stomach or chemotherapy after ESD<br>(iii) Discontinuation of anti-ulcer agents for any reason after ESD |
| Endoscopist performing ESD | Either an experienced endoscopist or a resident under the supervision of an experienced endoscopist | One of two board-certified endoscopists who had previously performed ESDs in over 100 gastric neoplasm cases |
| Instrument used to dissect gastric neoplasms | The insulated-tipped knife-2 (Olympus Medical Systems, Co. Ltd., Tokyo, Japan) or a Flush Knife(Fuji Film Medical, Tokyo, Japan) | The insulated-tipped knife-2 (Olympus Optical Co., Tokyo, Japan) |
| Coagulation of ulcers | VIO 300D, ERBE Elektromedizin, Tübingen, Germany | Haemostatic forceps (Coagrasper; Olympus Optical Co.) |

RCT, Randomised controlled trial; ESD, Endoscopic submucosal dissection; VPZ, Vonoprazan; PPI, Proton pump inhibitor; ECOG, Eastern Cooperative Oncology Group; Hb, hemoglobin; Plt, platelet count; AST, aspartate aminotransferase; ALT, alanine aminotransferase; Cre, Creatinine; PT, prothrombin time.

study is constant across strata [18, 19]. The details of the assumptions that the CDS estimator is unbiased are shown in the S1 and S2 Appendices. This study evaluated the effectiveness of vonoprazan in patients taking antithrombotic drugs, focusing on the exclusion criteria for antithrombotic drug use. Antithrombotic drugs were defined as taking either antiplatelet or anticoagulant drugs, and data based on the study by Kagawa et al. that included patients taking antithrombotic drugs were used for the observational study. A summary of the pooled studies is presented in Table 1. This study was approved by the Ethics Committee of the Graduate School and Faculty of Medicine, Kyoto University (approval number: R2141) and the Ethics Committee of Kagawa Prefectural Central Hospital (approval number: 839), and adhered to the Declaration of Helsinki and Ethical Guidelines for Medical and Health Research Involving Human Subjects.

## Randomized clinical trial

To identify studies eligible for analysis, a literature search was conducted using the PubMed, Scopus, and Cochrane library databases for all articles published through November 2020. The initial search terms included 'vonoprazan', 'Takecab', 'TAK-438', 'potassium-competitive inhibitor', 'ESD', and 'endoscopic submucosal dissection'. An example of a full electronic search strategy used for the online database is shown in S1 Table and a flow diagram for assessing the studies is shown in Fig 1.

A total of 101 articles were identified by literature search, and after letters, meta-analyses, and duplicate articles and protocols were excluded, six RCTs were eligible for CSD pooling.

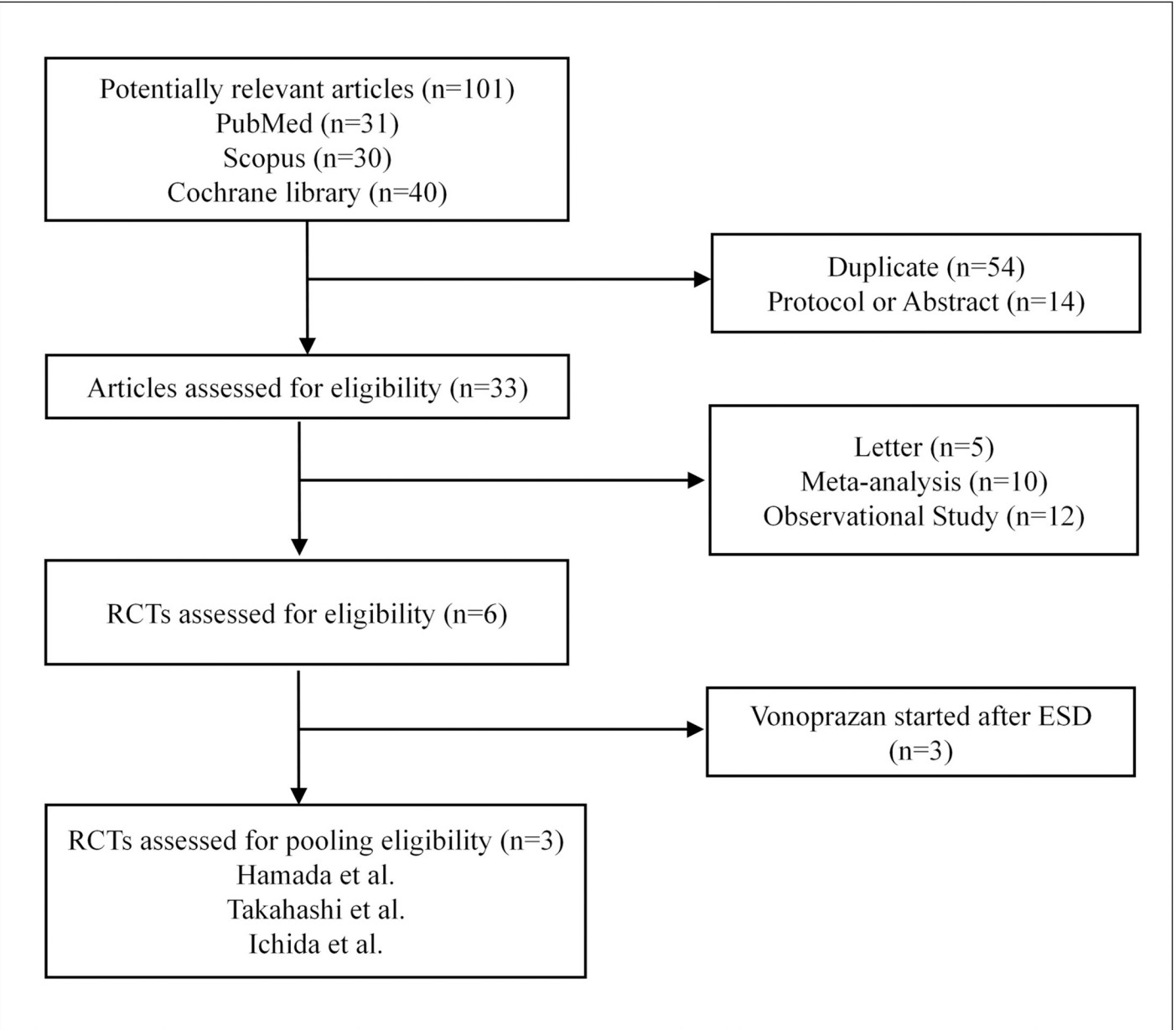

**Fig 1. Flow diagram of assessment of randomized controlled trials used in CDS.** RCTs, Randomised controlled trials; ESD, Endoscopic submucosal dissection.

All six RCTs compared the efficacy of vonoprazan with PPIs for post-ESD bleeding. Vonoprazan was administered 1 day before ESD in three studies, and a few days after ESD in the other three. The former group of RCTs was considered eligible for CDS because these studies used the same vonoprazan dosing regimen as the study by Kagawa et al. [9] (i.e., vonoprazan administration the day before ESD). We decided to use the RCT by Hamada et al., which set the primary endpoint as bleeding after ESD and calculated the planned sample size [10].

The study by Hamada et al. was an open-label, prospective, randomized phase II clinical trial conducted at the Osaka International Cancer Center in Japan [10]. One hundred and forty patients who underwent ESD between 2015 and 2016 were enrolled, and patients taking

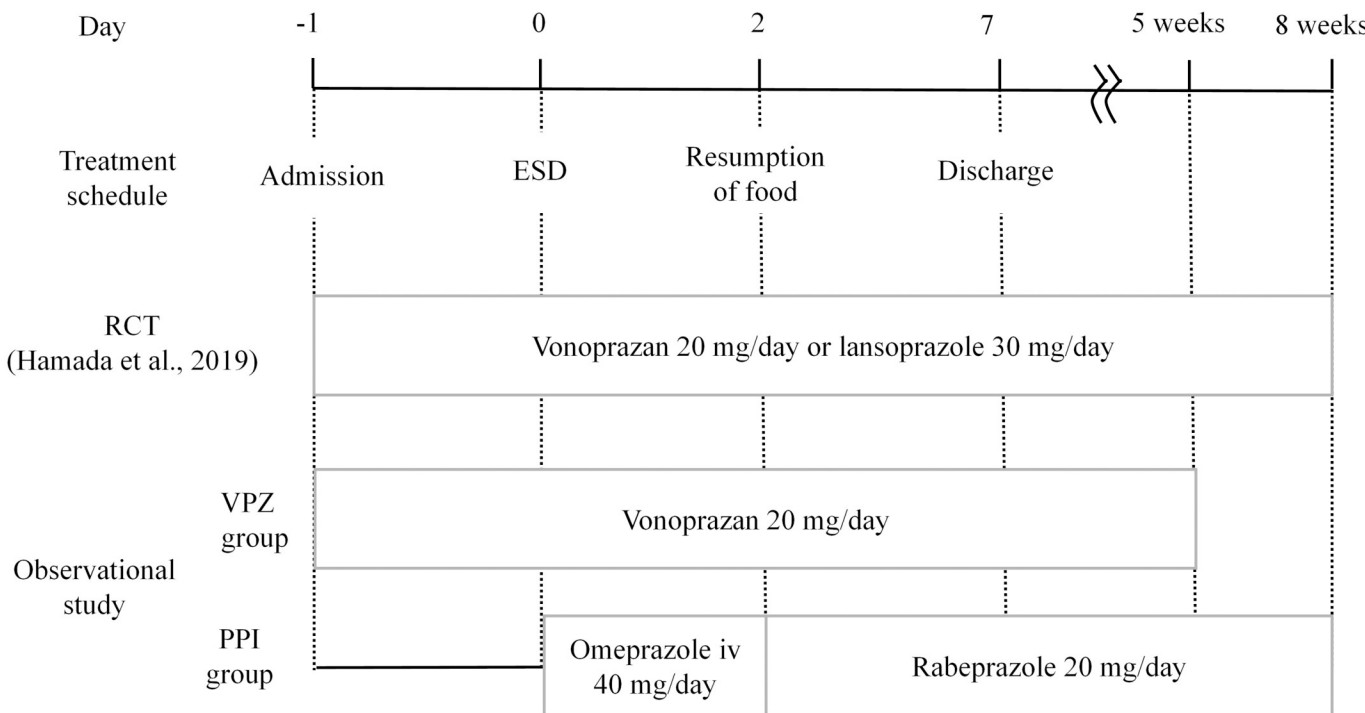

**Fig 2. Treatment schedules of the randomized clinical trial and observational study.** RCT, Randomised controlled trial; ESD, Endoscopic submucosal dissection; VPZ, Vonoprazan; PPI, Proton pump inhibitor; iv, Intravenous.

antithrombotic drugs were excluded. Patients were randomized in a 1:1 ratio to vonoprazan (n = 69) or PPI (n = 70).

## Observational study

For the observational study data, we used the data of ESD cases performed at our hospital from 2005 to 2018. The study by Kagawa et. al. used data up to 2015, but the sample size of 17 patients taking antithrombotic drugs was small and considered insufficient to examine the effect in patients taking antithrombotic drugs. Therefore, in this study, we used data up to 2018, when the sample size of antithrombotic and non-antithrombotic patients was more than 70 cases per group, similar to the sample size design of Hamada et al.

The observational study data analyzed in the CDS included patients taking antithrombotic drugs. In this study, 408 patients who underwent ESD and were treated with vonoprazan for postoperative ulcer treatment at Kagawa Prefectural Central Hospital in Japan from April 2014, when vonoprazan was launched in April 2018 were prospectively enrolled in the vonoprazan group. Then, from among 870 patients who underwent ESD and were treated with PPIs from 2005 to 2014, before the launch of vonoprazan, were analyzed as historical controls. All participants provided written informed consent, and they were given the opportunity to opt out of the present study.

## Treatment and follow-up

When performing CDS, the outcomes and study protocols of the studies to be pooled should be largely consistent. The treatment schedules in the RCT and observational studies are shown

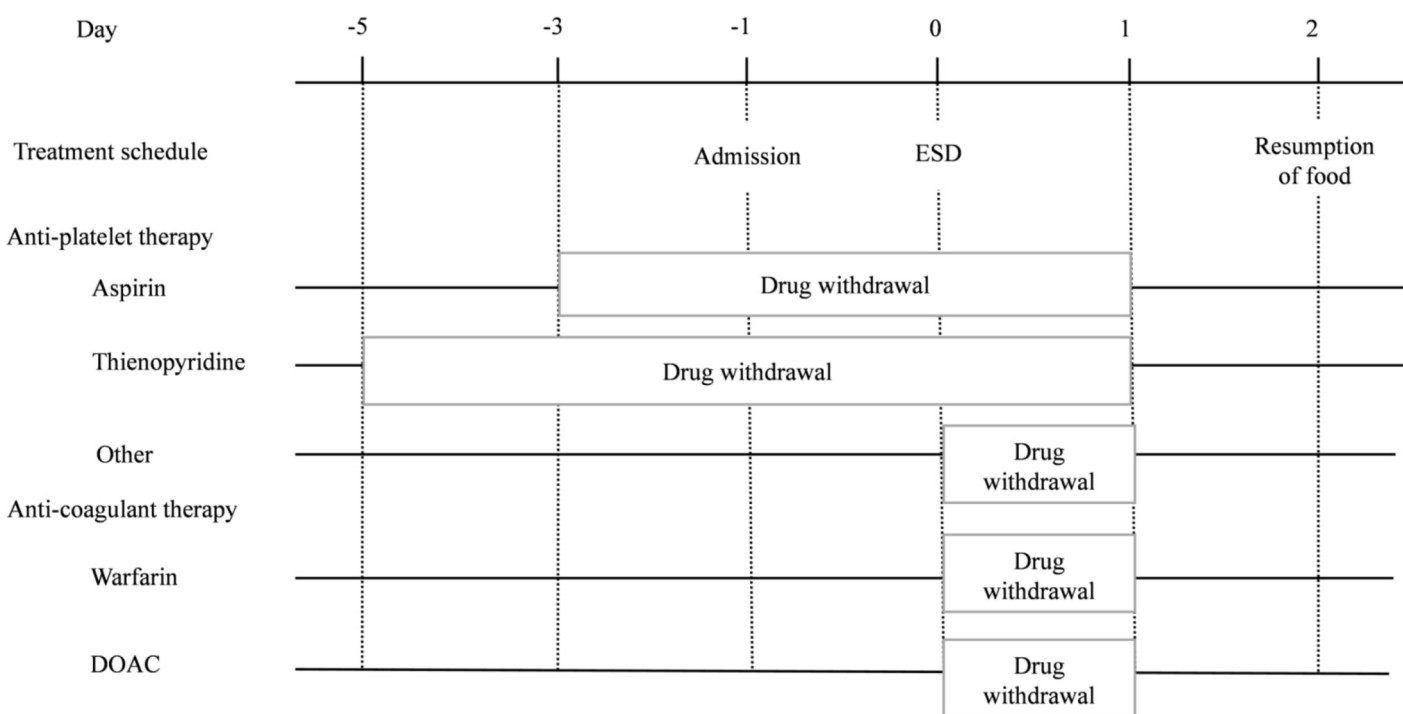

**Fig 3. The withdrawal schedules of each antithrombotic drug in the observational study.** ESD, Endoscopic submucosal dissection.

in Fig 2. The withdrawal schedules for each antithrombotic drug in the observational study are shown in Fig 3.

Participants in both studies were admitted the day before ESD, and all participants were hospitalized for 7 days. In both studies, the vonoprazan group started oral vonoprazan 20 mg on the day before ESD [9, 10]. The PPI group in the RCT started oral lansoprazole 30 mg the day before ESD, but those in the observational study received omeprazole intravenously for 2 days (day 0 and day 1, 40 mg/day) and then started oral rabeprazole (from day 2, 20 mg/day) [9, 10]. An infusion of omeprazole is known to rapidly raise the pH in the stomach to above 4.2 within a few hours [20]. On the other hand, oral PPIs require several days of dosing to show effects because they must be activated by acid [21, 22]. Therefore, the PPI groups in the two studies can be regarded as roughly equivalent regarding the PPI doses intended to produce an effect on the day of ESD.

Vonoprazan was administered for 8 weeks in the RCT and 5 weeks in the observational study. Most post-ESD bleeding events develop within the first two weeks after ESD [23]. In the study by Hamada et al., the Kaplan-Meier method was used to determine the cumulative non-bleeding rate within 28 days after ESD, and all bleeding occurred within 14 days after ESD [10]. Data from the observational study were used to analyze the outcomes up to 28 days after ESD. Thus, the durations of drug administration and outcome observations in the two studies were the same.

## Study outcome

The primary outcome of this study was post-ESD bleeding, which was defined as hematemesis, melena, or a decrease in hemoglobin of $\leq 2$ g/dl in both the RCT and observational studies [9, 10].

## Statistical analysis

The CDS in this study used the estimate of the treatment effect in patients not taking antithrombotic drugs, derived from the RCT, and the estimate of the treatment effect stratified by use of antithrombotic drugs, derived from the observational study. For the estimate from the RCT, we used the risk difference shown in Hamada et al. [10].

Propensity score matching (PSM) was performed to avoid the effects of measurable confounding factors when using observational study data. One-to-one matching without replacement was completed using the nearest neighbor match on the logit of the propensity score for the treatment approach with a caliper width set to 0.2 times the standard deviation of the logit of the propensity score. To estimate the propensity score, we fitted a logistic regression model for treatment as a function of patient characteristics, specifically age, *Helicobacter pylori* infection, tumor size, and tumor location. PSM was performed in the overall treated patients, patients taking antithrombotic drugs, and patients not taking antithrombotic drugs, and the risk difference in the matched data was calculated for each patient population.

Summary statistics were constructed using frequencies and proportions for categorical data and means and standard deviations for continuous data. Standardized differences were estimated for the factors used in the propensity score model after matching to assess post-match balance. Standardized differences of less than 10.0% for a given covariate indicate a relatively small imbalance [24]. The Wald confidence interval (CI) was used to indicate risk differences in the observational study. Confidence intervals for the CDS-based estimates were calculated using the method presented by Kaizer [18]. All statistical analyses were performed using SAS software (version 9.4; SAS Institute, Cary, NC, USA) and R version 3.6.3 (R Foundation for Statistical Computing, Vienna, Austria).

## Results

### Study population

We analyzed the RCT data from Hamada et al. comprising [10] 70 patients in the vonoprazan group and 69 in the PPI group along with the observational study data of 408 patients in the vonoprazan group and 870 in the PPI group. The patient characteristics from each study are shown in Table 2. With the use of PSM in the overall treated patients, 392 patients in the vonoprazan group were matched with 392 patients in the PPI group. Similarly, among the patients who were administered antithrombotic drugs, 86 patients in the vonoprazan group were matched with 86 patients in the PPI group. In addition, among patients not consuming

**Table 2. Patient characteristics in the randomized clinical trial and observational study before propensity score matching.**

| Characteristic | RCT (Hamada et al., 2018) | | Observational study | |
|---|---|---|---|---|
| | VPZ, n (%) | PPI, n (%) | VPZ, n (%) | PPI, n (%) |
| Total | 69 | 70 | 408 | 870 |
| Age, mean ± SD | 70.3 ± 6.8 | 70.1 ± 8.2 | 73.1 ± 8.9 | 71.4 ± 9.1 |
| Male | 51 (74%) | 57 (81%) | 295 (72.3%) | 646 (74.3%) |
| Positive for *Helicobacter pylori* | 27 (39%) | 29 (41%) | 142 (34.8%) | 366 (42.1%) |
| Endoscopic tumor size (mm), mean ± SD | 17 ± 13 | 16 ± 10 | 36.7 ± 13.4 | 35.4 ± 12.9 |
| Tumor size >2 cm | 12 (17%) | 11 (16%) | 376 (92.2%) | 813 (93.4%) |
| Tumor located in the antrum | 34 (49%) | 36 (51%) | 260 (63.7%) | 477 (54.8%) |
| Oral anti-thrombotic drug | - | - | 95 (23.3%) | 106 (12.2%) |

RCT, Randomised controlled trial; VPZ, Vonoprazan; PPI, Proton pump inhibitor; SD, Standard deviation.

**Table 3. Patient characteristics in the observational study after propensity score matching according to antithrombotic drug treatment.**

| Characteristic | Overall treated patients | | | Taking antithrombotic drugs | | | Not taking antithrombotic drugs | | |
|---|---|---|---|---|---|---|---|---|---|
| | VPZ, n(%) | PPI, n(%) | Standardized difference, % | VPZ, n(%) | PPI, n(%) | Standardized difference, % | VPZ, n(%) | PPI, n(%) | Sstandardized difference, % |
| Total | 392 | 392 | - | 86 | 86 | - | 306 | 306 | - |
| Age, mean ± SD | 72.6 ± 8.7 | 72.3 ± 8.8 | 2.6 | 75.3 ± 8.9 | 75.0 ± 9.1 | 3.5 | 72.1 ± 8.9 | 71.9 ± 9.1 | 1.9 |
| Positive for *Helicobacter pylori* | 142 (36.2%) | 142 (36.2%) | 0.0 | 35 (40.7%) | 38 (44.2%) | 7.1 | 105 (34.3%) | 108 (35.3%) | 2.1 |
| Tumor size >2 cm | 368 (93.9%) | 362(92.3%) | 6.0 | 84 (97.7%) | 84 (97.7%) | 0.0 | 286 (93.5%) | 285 (93.1%) | 7.9 |
| Tumor located in the antrum | 244 (62.2%) | 245 (62.5%) | 0.5 | 56 (65.1%) | 56 (65.1%) | 0.0 | 188 (61.4%) | 194 (63.4%) | 4.0 |

VPZ, Vonoprazan; PPI, Proton pump inhibitor; SD, Standard deviation.

antithrombotic drugs, 306 patients in the vonoprazan group were matched with 306 patients in the PPI group. Out of the total number of treated patients, 16 patients (nine patients taking antithrombotic medications and seven patients not taking antithrombotic medications) were dropped because of caliper matching. Among those that were matched, the standardized differences were below 10.0% for all factors, indicating only small differences between the two groups (Table 3).

## Primary outcome

The efficacy of vonoprazan compared with that of PPIs in preventing post-ESD bleeding is summarized in Table 4. Based on observational study data, the bleeding rate was 5.4% in the vonoprazan group and 5.6% in the PPI group. The estimated treatment effect among all the treated patients (the risk difference), including those taking and not taking antithrombotic drugs, was -0.3% (95% CI: -3.4% to −2.9%). In contrast, the estimate obtained by pooling data from the RCT and the observational study using CDS was -4.3% (95% CI: -11.7% to 3.1%). The bleeding rate in patients not taking antithrombotic medication was 4.3% in the RCT and 3.9% in the observational study for the vonoprazan group, 5.7% in the RCT, and 2.3% in the observational study for the PPI group. The estimated treatment effect was -1.4% (95% CI: -10.3% to 7.3%) in the RCT and 1.6% (95% CI: -1.1% to 4.4%) in the observational study. The bleeding rates among patients taking antithrombotics were 9.3% and 20.9% in the vonoprazan

**Table 4. Efficacy of vonoprazan compared with proton pump inhibitors in preventing post-ESD bleeding according to antithrombotic drug treatment.**

| | Post-ESD bleeding | | Risk difference (%) | 95% confidence interval |
|---|---|---|---|---|
| | VPZ, n (%) | PPI, n (%) | | |
| Overall treated patients | | | | |
| Observational study | 21/392 (5.4%) | 22/392 (5.6%) | -0.3 | -3.4 to 2.9 |
| Cross-design synthesis | - | | -4.3 | -11.7 to 3.1 |
| Patients not taking antithrombotic drugs | | | | |
| RCT | 3/69 (4.3%) | 4/70 (5.7%) | -1.4 | -10.3 to 7.3 |
| Observational study | 12/306 (3.9%) | 7/306 (2.3%) | 1.6 | -1.1 to 4.4 |
| Patients taking antithrombotic drugs | | | | |
| Observational study | 8/86 (9.3%) | 18/86 (20.9%) | -11.6 | -22.2 to -1.1 |
| Cross-design synthesis | - | | -14.6 | -22.0 to -7.2 |

ESD, Endoscopic submucosal dissection; VPZ, Vonoprazan; PPI, Proton pump inhibitor; RCT, Randomised controlled trial

**Table 5. Post-ESD bleeding by types of antithrombotic drugs taken.**

|  | VPZ, n (%) | PPI, n (%) | Risk difference (%) | 95% confidence interval |
|---|---|---|---|---|
| Total | 8 / 86 (9.3%) | 18 / 86 (20.9%) | -11.6 | -22.2 to -1.1 |
| Anti-platelet monotherapy | 2 / 45 (4.4%) | 3 / 37 (8.1%) | -3.7 | -14.3 to 7.0 |
| Dual anti-platelet agents | 2 / 26 (7.7%) | 7 / 24 (29.2%) | -21.5 | -42.4 to -0.01 |
| Anti-coagulant monotherapy (warfarin) | 3 / 8 (37.5%) | 3 / 17 (17.6%) | 19.9 | -18.3 to 57.9 |
| Anti-coagulant monotherapy (DOAC) | 0 / 3 (0.0%) | - / 0 | - | - |
| Combination of anti-platelet and anti-coagulant (warfarin) | 0 / 3 (0.0%) | 4 / 7 (57.1%) | -57.1 | -93.8 to -20.5 |
| Combination of anti-platelet and anti-coagulant (DOAC) | 1 / 2 (50.0%) | 1 / 1 (100.0%) | -50.0 | -100.0 to 19.3 |

ESD, Endoscopic submucosal dissection; VPZ, Vonoprazan; PPI, Proton pump inhibitor; DOAC, Direct oral anticoagulant

and PPI groups, respectively. The estimated treatment effect was -11.6% (95% CI: -22.2% to -1.1%) based on the observational study; however, the estimate obtained using CDS was -14.6% (95% CI: -22.0% to -7.2%).

The results of post-ESD bleeding according to the type of antithrombotic drug used are shown in Table 5. Of the 86 patients in the vonoprazan group, 45 were on antiplatelet mono-therapy, such as low-dose aspirin (LA)—32 were on LA, 11 on thienopridine, and 2 on other antiplatelet agents. Twenty-six patients received dual antiplatelet therapy (four received LA and thienopridine, seven received LA and other antiplatelet agents, and 15 received two other antiplatelet agents), of which two patients suffered post-ESD bleeding. Eight patients were on anticoagulant monotherapy (warfarin) and three were on direct oral anticoagulants (DOACs) (two on rivaroxaban and one on dabigatran), of which three and zero patients, respectively, experienced post-ESD bleeding. Three patients were on both, an antiplatelet agent and an anti-coagulant (warfarin) (one on LA and warfarin, two on other antiplatelet agents and warfarin) and two patients were on both, an antiplatelet agent and an anticoagulant (DOAC) (one on other antiplatelet agents and apixaban, one on LA and dabigatran), of which zero and one patient, respectively, showed post-ESD bleeding. Of the 86 patients in the PPI group, 37 were on antiplatelet monotherapy (23 on LA, 10 on thienopridine, 4 on other antiplatelet agents) and 24 patients were on dual antiplatelet therapy (4 on LA and thienopridine, 7 on LA and other antiplatelet agents, 12 on two other antiplatelet agents), of which 3 and 7 patients, respectively, experienced post-ESD bleeding. Seventeen patients were on anticoagulant monotherapy (warfarin), of which three patients suffered post-ESD bleeding. Seven patients were on both, an antiplatelet agent and an anticoagulant (warfarin) (four on LA and warfarin, three on other antiplatelet agents and warfarin) and one patient was on both, an antiplatelet agent and an anticoagulant (DOAC) (LA and rivaroxaban), of which four and one patient, respectively, experienced post-ESD bleeding.

## Discussion

An observational study by Kagawa et al. [9] suggested that vonoprazan significantly reduced post-ESD bleeding compared to PPIs. However, these results were inconsistent with the results of the RCT conducted by Hamada et al. [10], which led us to perform the present study.

In their observational study, Kagawa et al. revealed that the risk difference of post-ESD bleeding in patients consuming antithrombotic drugs between vonoprazan and PPIs was -36.7%; however, the sample size for patients taking antithrombotic drugs was insufficient. In addition, the robustness of evidence from observational studies is limited. Therefore, in the present study, we increased the sample size of patients receiving antithrombotic drugs and

examined the results using the CDS method. This guided treatment selection bias in the observational study data. Subsequently, the risk difference in patients taking antithrombotics was -11.6%, while that using the CDS method was -14.6%. Regarding tumor size, the patients enrolled in the observational study had a larger tumor size than those enrolled in the RCT. Therefore, the estimates of treatment effects from observational study data may be biased due to higher bleeding rates. However, since the percentage of patients with large tumors was similar between patients with and without antithrombotic medication in the observational study, the CDS estimator may have excluded such bias, by the definition of sufficient conditions under which the CDS estimator is unbiased.

The two main types of anticoagulants are warfarin and DOACs. In two studies consisting of a small number of DOAC users, the frequency of post-ESD bleeding events was reported to be 20.8% [25] and 22.0% [26], respectively. The risk of post-ESD bleeding was reportedly higher with warfarin than with DOACs in high-risk endoscopic procedures. However, there was no difference in the risk of bleeding during upper gastrointestinal ESD [27]. In addition, the risk of post-ESD bleeding differs between individual DOAC users [25]. Since only six DOAC users were included in this study, further studies are needed to determine the risk of post-ESD bleeding with specific anticoagulant drugs.

In this study, most of the enrolled antithrombotic drug users were receiving a single antiplatelet agent. A previous meta-analysis revealed that single antiplatelet agent use was also a risk factor for post-ESD bleeding, with an odds ratio of 2.08 compared to non-users [17]. Other studies have shown that the odds ratios for post-ESD bleeding in patients who received double antiplatelet therapy [28], warfarin [29], or LDA and warfarin [30] were 5.06, 10.15, and 14.38, respectively, when compared with non-antithrombotic drug users. In this study, following PSM, we assessed the difference in the incidence of post-ESD bleeding between vonoprazan and PPIs according to the type of antithrombotic drug used. The post-ESD bleeding rate with the combined use of an antiplatelet and anticoagulant (warfarin or DOAC), which are presumed to pose the highest risk of bleeding, was lower in the vonoprazan group than in the PPI group (20.0% vs. 62.5%). The superiority of vonoprazan over PPI in preventing post-ESD bleeding may, therefore, be more pronounced in patients at a high risk of bleeding.

Several attempts have been made to prevent post-ESD bleeding. Long-term discontinuation of antithrombotic drugs before and after gastric ESD may help avoid increasing the risk of post-gastric ESD bleeding caused by antithrombotic drugs. Several retrospective studies have reported that discontinuing antithrombotic therapy a week before gastric ESD does not increase the risk of thromboembolism [31, 32]. However, other studies have shown that prolonged discontinuation can increase the risk of thromboembolism [33, 34]. Traditionally, bridging therapy from oral warfarin to intravenous heparin has been used during the perioperative period of ESD, considering both thromboembolism and post-ESD bleeding. Unfortunately, bridging therapy in anticoagulant users is not appropriate for gastric ESD because it does not decrease every complication [25, 27]. Second-look endoscopy (SLE) has been performed after ESD to evaluate the ulcer condition and to perform preventive hemostatic treatment for high-risk bleeding ulcers [35]. However, SLE cannot suppress bleeding after ESD, and bleeding prevention procedures performed during SLE may be harmful [36, 37]. Therefore, other methods are necessary to suppress post-ESD bleeding in antithrombotic drug users.

In upper gastrointestinal bleeding, the intragastric pH significantly affects the coagulation system (including platelets), which is involved in hemostasis. An intragastric pH of 5.4 or higher is required for blood coagulation to achieve hemostasis in peptic ulcers [38] because pepsin causes platelet destruction at pH 5.0–5.5 [39]. This suggests that maintaining an intragastric pH of at least 5.4 is extremely important for suppressing post-ESD bleeding. Treatment

for ESD-induced artificial ulcers with gastric acid inhibitors is usually initiated on the date of ESD. However, post-ESD bleeding occurs most frequently within 24 hours after the procedure [5, 35]. In the observational study analyzed here, vonoprazan was administered 1 day before ESD to achieve a high intragastric pH on the day of performing the technique, thereby facilitating effective hemostatic function. The RCT analyzed in the present study also involved the administration of vonoprazan in the same way.

Concerning gastric acid suppression, PPI administration can also achieve an intragastric pH of 5 or more, even when administered intravenously or orally. In the observational study, a combination of 2-day intravenous infusions of omeprazole and an oral intake of 20 mg of rabeprazole was adopted in the PPI group. A single infusion of omeprazole is known to quickly increase the intragastric pH >4.25 [20]. On the other hand, for the PPI group in the RCT, administration of oral lansoprazole 30 mg was started the day before ESD was conducted. The ratios of post-ESD bleeding in non-antithrombotic drug users were low and almost the same in the observational study and RCT (2.3% vs. 5.7%, respectively), suggesting that a sufficiently high intragastric pH was achieved by PPI administration on the day of ESD, to provide adequate prophylaxis against bleeding in non-high-risk patients.

A crossover study compared vonoprazan with a PPI in terms of inhibiting gastric acid secretion. The holding time ratios at an intragastric pH of 5 or higher on day 1 after administration of vonoprazan and esomeprazole were 62.8% and 13.0%, respectively, and those for vonoprazan and rabeprazole were 76.6% and 16.7%, respectively. However, esomeprazole and rabeprazole were better able to achieve an intragastric pH of 5 or higher after 7 days of administration (48.3% and 53.2%, respectively) [8]. Although initiating PPI therapy 7 days before ESD is difficult in everyday clinical practice, it might further reduce the risk of bleeding in antithrombotic drug users.

Because of various types of biases, patients with medical backgrounds that might affect the results were excluded from the study. However, it has been reported that the results of RCTs obtained in this way differ from those observed in real-world clinical practice [40–42]. The CDS method was designed by the United States General Accounting Office to address the generalizability problem caused by sample selection bias in RCTs [43]. Recently, the statistical properties of the CDS estimator have been evaluated by Kaizer [18]. In the simulation studies, the CDS estimates had a lower bias than the commonly used estimates based on randomized or observational studies alone under reasonable data assumptions. Therefore, very few studies have employed CDS to analyze real-world clinical data [18, 44]. In this study, CDS analysis suggested that vonoprazan is more effective than PPIs in preventing post-ESD bleeding in patients taking antithrombotic drugs. It may be difficult to perform RCTs targeting antithrombotic drug users with various comorbidities. However, we believe that future observational studies will confirm the validity of the results obtained in this study.

This study had several limitations. First, even if the confounding effects could have been examined by CDS, selection bias cannot be ruled out because both RCT and observational studies were from a single center. CDS estimates may depend on the data used; therefore, it is necessary to examine a variety of data. In addition, when integrating data from two studies, all items were not the same in the design of the observational studies and RCTs. Second, the CDS method cannot account for all unknown confounders, and the validity of sufficient conditions cannot be confirmed from the data. Therefore, the bias of the estimates using CDS is not always zero, and further improvement of the method is desired. Third, although the same experienced endoscopists performed ESD in the PPI and vonoprazan groups in the observational study, the results may have been influenced by the endoscopists' experience. Finally, neither of the two studies analyzed here investigated the relationship between gastric pH and post-ESD bleeding.

## Supporting information

**S1 Table. Example of full electronic search strategy used for the online database.**
(PDF)

**S1 Appendix. The CDS estimator.**
(PDF)

**S2 Appendix. Sufficient assumptions that the CDS estimator is unbiased.**
(PDF)

## Author Contributions

**Conceptualization:** Yu Hidaka, Toru Imai, Tomoki Inaba, Tomo Kagawa, Katsuhiro Omae, Shiro Tanaka.

**Data curation:** Yu Hidaka, Tomo Kagawa.

**Formal analysis:** Yu Hidaka.

**Funding acquisition:** Yu Hidaka.

**Investigation:** Yu Hidaka, Toru Imai, Tomoki Inaba.

**Methodology:** Yu Hidaka, Toru Imai, Tomoki Inaba, Tomo Kagawa, Katsuhiro Omae, Shiro Tanaka.

**Project administration:** Yu Hidaka, Toru Imai, Tomoki Inaba, Tomo Kagawa.

**Resources:** Yu Hidaka, Tomoki Inaba, Tomo Kagawa.

**Software:** Yu Hidaka.

**Supervision:** Toru Imai, Tomoki Inaba, Tomo Kagawa, Katsuhiro Omae, Shiro Tanaka.

**Validation:** Yu Hidaka, Tomoki Inaba.

**Visualization:** Yu Hidaka, Tomoki Inaba.

**Writing – original draft:** Yu Hidaka, Tomoki Inaba, Tomo Kagawa.

**Writing – review & editing:** Yu Hidaka, Toru Imai, Tomoki Inaba, Tomo Kagawa, Katsuhiro Omae, Shiro Tanaka.

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
