## [Decision Letter · Decision Letter 0]

1 Oct 2021

PONE-D-21-19373Efficacy of vonoprazan against bleeding from endoscopic submucosal dissection-induced gastric ulcers under antithrombotic medication: A cross-design synthesis of randomized and observational studiesPLOS ONE

Dear Dr. Hidaka,

Thank you for submitting your manuscript to PLOS ONE. After careful consideration, we feel that it has merit but does not fully meet PLOS ONE’s publication criteria as it currently stands. Therefore, we invite you to submit a revised version of the manuscript that addresses the points raised during the review process.

We look forward to receiving your revised manuscript.

Kind regards,

Sanjiv Mahadeva, MRCP, MD

Academic Editor

PLOS ONE

2. Please ensure you have provided the full electronic search strategy for at least one database

4. Thank you for stating the following in the Acknowledgments/ Funding Section of your manuscript:

“This study was funded by AMED under Grant Numbers JP19lk0201061 and 340

JP20lk0201061.”

Additional Editor Comments (if provided):

The study is of clinical interest, but the manuscript needs some revision. Please respond to the reviewers' comments. The Discussion needs to be simplified as well.

Reviewers' comments:

Reviewer's Responses to Questions

**Comments to the Author**

1. Is the manuscript technically sound, and do the data support the conclusions?

Reviewer #1: Partly

Reviewer #2: Partly

2. Has the statistical analysis been performed appropriately and rigorously? 

Reviewer #1: N/A

Reviewer #2: Yes

3. Have the authors made all data underlying the findings in their manuscript fully available?

Reviewer #1: Yes

Reviewer #2: Yes

4. Is the manuscript presented in an intelligible fashion and written in standard English?

Reviewer #1: Yes

Reviewer #2: Yes

5. Review Comments to the Author

Reviewer #1: Dear Editors and authors,

I have read the manuscript PONE-D-21-19373 entitled “Efficacy of vonoprazan against bleeding from endoscopic submucosal dissectioninduced gastric ulcers under antithrombotic medication: A cross-design synthesis of randomized and observational studies” with interest. The authors used the uncommon but interesting statistic method in data analysis to evaluate the efficacy of vonoprazan in preventing post ESD bleeding in patients taking anti-platelet and/or anti-coagulant, compared to PPI. However, there are many important major concerns that need to clarify in the manuscript as the followings;

Major concerns

1. From table 5, the percentage of combination of anti-platelet and anti-coagulant (warfarin) was obviously higher in PPI group (0% vs. 57%). This maybe the factor contributing higher rebleeding rate in PPI group. Please discuss on this issue.

2. In discussion part, the author stated that “In patients taking antithrombotic drugs, vonoprazan was expected to reduce post-ESD bleeding to a greater extent than PPIs, and the observational study by Kagawa et al. [7] suggested that vonoprazan reduced post-ESD bleeding in patients taking antithrombotic drugs. However, these results were inconsistent with those of the subsequent RCT by Hamada et al. [8], which led us to perform the present study.” In fact, the study by Hamada excluded the patients taking antithrombotic drugs, thus the results were different from previous observational study as they assessed the patients in different group. Please consider re-write.

3. The authors said that “An intragastric pH of 5.4 or higher is required for blood coagulation to achieve haemostasis in peptic ulcer” and showed that Vonoprazan provided the targeted pH. However, this was not only the ability of Vonoprazan but also PPIs. Please revised those sentences, otherwise the reader maybe confused.

4. Please put the reference to this sentence “It remains unanswered whether the results obtained in RCTs are inconsistent with the facts observed in real-world clinical practice, which involves treating patients from diverse backgrounds.” In theory, this sentence may not true as the real-world data contains various type of bias and that is the reason for conducting RCT.

5. The discussion part was too long and some sentences eg. “Keith and Constance reported that clinical studies evaluating the effects of alcohol treatment excluded patients with more severe alcohol disorders and those with low incomes and psychiatric problems”, etc. were not relevant to the present study. Please consider shortening the discussion part.

Reviewer #2: Hidaka et al. conducted a cross-design synthesis evaluating the treatment effects between Vonoprazan and Proton-pump inhibitor in anti-thrombotic users who underwent gastric ESD. They concluded that from the pooled data of selected from one RCT and one observational data from their center as a historical control, Vonoprazan is more effective than PPI in preventing post-ESD bleeding among patients using anti-thrombotics. The study is clinically relevant with an interesting concept. However, a few shortcomings exist.

1. The definition of “taking anti-thrombotic medication” lacks the timing and duration in relation to the procedure i.e. were the medications taken throughout periprocedural period or were they stopped at any point prior to the procedure.

2. Among 3 RCTs included, the other two studies had sample size calculated too. Please specify more elaborative reasons why the study by Hamada et al. was chosen over the other two.

3. Since Vonoprazan requires shorter onset of action and shorter duration to reach its steady level, administering oral PPI one day prior to the ESD procedures may not be adequate time for bleeding prophylaxis. This limitation should be discussed.

4. As ESD with PPI prophylaxis performed before Vonoprazan became available were used as historical control, the bleeding outcome is inevitably confounded by endoscopists' experience which would be less than ESDs performed later (Vonoprazan group). Please discuss this limitation.

5. Statistically significant difference between each type of antithrombotic medications in each group (PPI vs Vonoprazan) should be presented in Table 5.

6. The authors rightfully stated that the risk of post-ESD bleeding differed between individual DOACs and other antithrombotic medication therefore, the differences between each antithrombotic regimen in both Vonoprazan and PPI group in this study needs to be further elaborated.

7. There is a few numerical errors e.g. Specific anti-thrombotic stratification is presented in Table 5, not 4

8. The difference in pharmacokinetic of vonoprazan and PPI should be moved to introduction

9. There are a few grammatical and spelling errors. Please proofread the manuscript again.

6. PLOS authors have the option to publish the peer review history of their article (what does this mean?). If published, this will include your full peer review and any attached files.

Reviewer #1: No

Reviewer #2: No

---

## [Author Response · Author response to Decision Letter 0]

3 Nov 2021

We thank the reviewers for their careful reading of our manuscript and their constructive criticism. The response to the reviewer has been written in the reviewer to response file.

---

## [Decision Letter · Decision Letter 1]

9 Dec 2021

Efficacy of vonoprazan against bleeding from endoscopic submucosal dissection-induced gastric ulcers under antithrombotic medication: A cross-design synthesis of randomized and observational studies

PONE-D-21-19373R1

Dear Dr. Hidaka,

We’re pleased to inform you that your manuscript has been judged scientifically suitable for publication and will be formally accepted for publication once it meets all outstanding technical requirements.

Kind regards,

Sanjiv Mahadeva, MRCP, MD

Academic Editor

PLOS ONE

Additional Editor Comments (optional):

The revised manuscript is satisfactory

Reviewers' comments:

Reviewer's Responses to Questions

**Comments to the Author**

1. If the authors have adequately addressed your comments raised in a previous round of review and you feel that this manuscript is now acceptable for publication, you may indicate that here to bypass the “Comments to the Author” section, enter your conflict of interest statement in the “Confidential to Editor” section, and submit your "Accept" recommendation.

Reviewer #1: All comments have been addressed

2. Is the manuscript technically sound, and do the data support the conclusions?

Reviewer #1: Yes

3. Has the statistical analysis been performed appropriately and rigorously? 

Reviewer #1: N/A

4. Have the authors made all data underlying the findings in their manuscript fully available?

Reviewer #1: Yes

5. Is the manuscript presented in an intelligible fashion and written in standard English?

Reviewer #1: Yes

6. Review Comments to the Author

Reviewer #1: The authors did response to the comments properly. The manuscript is well written. I have no further comments for this article.

7. PLOS authors have the option to publish the peer review history of their article (what does this mean?). If published, this will include your full peer review and any attached files.

Reviewer #1: No

---

## [Editor Report · Acceptance letter]

14 Dec 2021

PONE-D-21-19373R1 

Efficacy of vonoprazan against bleeding from endoscopic submucosal dissection-induced gastric ulcers under antithrombotic medication: A cross-design synthesis of randomized and observational studies 

Dear Dr. Hidaka:

I'm pleased to inform you that your manuscript has been deemed suitable for publication in PLOS ONE. Congratulations! Your manuscript is now with our production department. 

Kind regards, 

on behalf of

Dr. Sanjiv Mahadeva 

Academic Editor

PLOS ONE